# G2PDeep-v2: A Web-Based Deep-Learning Framework for Phenotype Prediction and Biomarker Discovery for All Organisms Using Multi-Omics Data

**DOI:** 10.3390/biom15121673

**Published:** 2025-12-01

**Authors:** Shuai Zeng, Trinath Adusumilli, Sania Zafar Awan, Manish Sridhar Immadi, Dong Xu, Trupti Joshi

**Affiliations:** 1Department of Electrical Engineering and Computer Science, University of Missouri, Columbia, MO 65211, USA; zengs@missouri.edu (S.Z.); tafbr@missouri.edu (T.A.); mizy9@missouri.edu (M.S.I.); xudong@missouri.edu (D.X.); 2Christopher S. Bond Life Sciences Center, University of Missouri, Columbia, MO 65211, USA; 3Institute for Data Science and Informatics, University of Missouri, Columbia, MO 65211, USA; sah2p@missouri.edu; 4Department of Biomedical Informatics, Biostatistics and Medical Epidemiology, University of Missouri, Columbia, MO 65211, USA; 5Department of Biomedical Sciences, Joan C. Edwards School of Medicine, Marshall University, Huntington, WV 25703, USA

**Keywords:** multi-omics, biomarker, phenotype prediction, deep learning, automated hyperparameters tunning, reproducibility, web-platform

## Abstract

Multi-omics data offers rich insights into complex traits across organisms, yet integrating and analyzing these datasets for phenotype prediction and marker discovery remains challenging. Researchers need accessible tools that combine deep learning, hyperparameter optimization, visualization, and downstream analysis in a unified web platform. To address this, we developed G2PDeep-v2, a web-based platform powered by deep learning for phenotype prediction and marker discovery from multi-omics data across a wide range of organisms, including humans and plants. The server provides multiple services for researchers to create deep-learning models through an interactive interface and train these models using an automated hyperparameter tuning algorithm on high-performance computing resources. Users can visualize the results of phenotype and markers predictions and perform Gene Set Enrichment Analysis for the significant markers to provide insights into the molecular mechanisms underlying complex diseases, conditions and other biological phenotypes being studied.

## 1. Introduction

With the advances in molecular profiling technologies, the ability to observe large-scale multi-omics data from patients or other biological organisms has grown remarkably over the past decade. Genome-wide data encompassing various molecular processes, such as gene expression, microRNA (miRNA) expression, protein expression, DNA methylation, single nucleotide polymorphisms (SNP), and copy number variations (CNV), can be obtained for the same set of samples, resulting in multi-omics data for numerous disease and crop studies. Although each type of multi-omics data captures a portion of the biological information, integrating multi-omics data helps researchers comprehensively understand biological systems from different perspectives [1,2]. Researchers have utilized multi-omics data to address many significant breeding and biomedical problems, including plant breeding [3], drug target discovery [4], disease therapy [5,6], and survival analysis. Specifically, muti-omics data allows researchers to predict the phenotypes and identify biomarkers that affect the diversity of phenotypes. To effectively take advantage of complementary information in multi-omics data, it is important to have a one-stop-shop platform for researchers to integrate multi-omics data, train customized deep-learning models for predicting phenotypes using high-performance computing resources and discover potential biomarkers along with their biological relevance.

Many approaches have been proposed over the past decade to utilize one type of omics data analysis for various bioinformatics problems. Early attempts have employed supervised learning methods for biomedical classification tasks. For example, DeepGS [7] applies a deep convolutional neural network combined with a fully connected neural network to predict phenotype based on SNP. Blaise et al. [8] proposed an approach for the biological interpretation of deep learning models for phenotype prediction from gene expression data. However, these methods only consider one of the multi-omics data types and fail to utilize useful biological information from other types of multi-omics data. Recently, more supervised methods focused on exploiting the interactions across different omics data types for better prediction. MOGONET [9] integrates multi-omics data using graph convolutional networks for biomedical classification tasks such as Alzheimer’s disease patient classification and kidney cancer type classification. Sammut et al. [10] introduced an ensemble-based machine learning framework to integrate representations from different multi-omics data types for breast cancer therapy response. Some efforts focus on biologically informed deep learning models with multi-omics data to enhance the interpretability of models [11,12,13].

Although these methods have shown some good performance, there are still challenges in adopting such models in different types of studies. The models used in these methods are typically designed for a specific study with a particular set of data, which means that researchers must invest considerable effort to adapt the model for other studies, as they are not generalizable. Inappropriate hyperparameter optimization is a common issue, which often negatively affects the performance of model and analytical outcomes. In other words, manually tuning the optimal hyperparameters is challenging due to the vast number of possible combinations. These methods have steep learning curves and often require complicated installation steps. Furthermore, training models with large-scale multi-omics data requires computing resources and storage exceeding the capacities of most potential non-computer savvy users. Moreover, few existing methods integrate functionalities to identify significant multi-omics signatures and biomarkers related to biomedical and biological studies, resulting in researchers spending additional time on confirming evidence for the findings.

Along this line of research, we have been developing the deep learning method G2PDeep. The first original v1 model was made available in 2019 [14], followed by the web server published in 2021 [15]. In its first version, G2PDeep enabled the quantitative phenotype prediction and marker discovery by using a dual-CNN model trained from scratch using only SNP. This work has gained a lot of interest from researchers worldwide, with more than 500 submissions for model training conducted via web-based access. To address the limitations discussed above, we have further expanded it to G2PDeep-v2, a comprehensive web-based platform for phenotype prediction using multi-omics data and biomarkers discovery for all organisms. Unlike the previous version of G2PDeep, the new version, G2PDeep-v2, now supports multiple inputs for multi-omics data, offers a broader array of model selection options, advanced settings for tuning model hyperparameters, and includes comprehensive Gene Set Enrichment Analysis (GSEA) functionalities. The difference between the previous and the new version of G2PDeep is clearly depicted in Table 1. Precisely, compared with other available applications, G2PDeep-v2 provides end-to-end management of machine learning projects from multi-omics dataset creation through to model interpretation, which also supports individual omics or any combination of up to three multi-omics data for the predictions. It is equipped with a fully automated pipeline to process and organize multi-omics data such as gene expression, miRNA expression, DNA methylation, protein expression SNP, and CNV. It provides an interactive web interface enabling machine learning and deep learning models to be created, and customized predictions according to different research tasks. It also provides automated hyperparameters search with Bayesian optimization algorithm, discovering a top-performing model configuration from a huge number of combinations of hyperparameters, without any manual effort necessary beyond just the initial set-up. It supports real time monitoring for ongoing model training and optimization history through a real-time web dashboard. The G2PDeep-v2 server is publicly available at https://g2pdeep.org/ and can be utilized for all organisms.

The datasets and well-trained models are serialized and stored in user accounts to protect the privacy of research information from unauthorized parties. The well-trained models can be retrieved from a pool of models to predict the phenotype and discover the significant biomarkers associated with the phenotype, making the models reusable and reproducible. The predicted results of phenotype are summarized in an interactive figure, and its raw results can be downloaded as a comma-separated values (CSV) file. The GSEA can be performed using significant biomarkers, Kyoto Encyclopedia of Genes and Genomes (KEGG) [16] and Reactome [17] pathway information, providing insights into pathways underlying the phenotype. The publications strongly associated with significant biomarkers in phenotype of user’s interest are listed in a table along with their abstracts and URL links, identifying the newest evidence from relevant research for the researchers.

Here, we present our multi-omics datasets exemplar studies for 23 different cancer with long-term-survival labels, originally provided by The Cancer Genome Atlas (TCGA) project [18] for biomedical applications and Soybean Cyst Nematode (SCN) resistance prediction in soybean for agribiotech application. We have utilized G2PDeep-v2 to train models with automating hyperparameters search on different combinations of multi-omics data and identified multiple sets of significant biomarkers. All these datasets, models, biomarkers with GSEA results are retrievable for all users and visitors. We demonstrated that G2PDeep-v2 can identify significant biomarkers associated with patient outcomes, as well as markers linked to resistance traits. To the best of our knowledge, G2PDeep-v2 is the only end-to-end, web-based deep-learning framework that supports phenotype prediction, biomarker discovery, and functional annotation from multi-omics data across diverse organisms, lowering the barriers for utilization and application of deep learning techniques, especially for non-informatics and computer savvy users, new to such techniques. Users can apply G2PDeep-v2 not only to human disease studies but also to other organisms including research in plants. The G2PDeep-v2 server is publicly available at https://g2pdeep.org (accessed on 1 August 2023).

## 2. Materials and Methods

### 2.1. Data Pre-Processing

To enhance the scalability of the dataset, G2PDeep-v2 employs one-hot encoding and normalization individually on six different types of omics data: gene expression, miRNA expression, DNA methylation, protein expression, SNP, and CNV. Regarding features in expression data, such as gene expression, miRNA expression, DNA methylation, and protein expression, the values in each sample undergo normalization through *z*-score normalization. Focusing on DNA methylation data, only CpG islands occurring in promoter regions or genes are included. For SNP data, the four genotypes (adenine (A), thymine (T), cytosine (C), and guanine (G)) and missing data undergo encoding through one-hot binary encoding. In the case of gene-level CNV data, the encoding includes homozygous deletion, single copy deletion, diploid normal copy, low-level copy number amplification, and high-level copy number amplification, utilizing one-hot binary encoding. Notably, missing values for expression data are set to 0, while none of the SNP and CNV datasets undergo any imputation process. For each input modality, G2PDeep-v2 automatically verifies the data format to ensure compatibility with the platform. This includes checking feature types, value ranges, and dimensional consistency. The system provides informative feedback if any discrepancies are detected, preventing misclassification of data types or incorrect combination of different omics modalities, and ensuring that integration and model training are performed accurately.

### 2.2. Modeling in G2PDeep

#### 2.2.1. Multi-CNN

Our proposed multi-CNN is an extended version of the dual-CNN reported in our previous work [14,15]. The multi-CNN model (as shown in Figure 1) takes up to three types of omics data combinations as input. The model consists of multiple parallel CNN layers and a fully connected neural network. The encoded genotypes for each type of omics are individually passed into multiple parallel CNN layers. These layers generate representations for each type of omics data to discover patterns and provide a better understanding of the biomarkers. The representations for each type of omics are concatenated, integrating the information of biomarkers from different perspectives. The concatenated representations are then passed into the fully connected neural network with an output layer for phenotype prediction. To prevent the model overfitting, a Batch Normalization [19] layer is added at the end of representation and Dropout [20] layers are added in each layer of fully connected neural network. The Leaky Rectified Linear Unit (Leaky-ReLU) [21] activation function is added to each layer of model. The loss function of the model is cross-entropy and mean squared error for categorical phenotype and quantitative phenotype prediction, respectively. The model is optimized by Adam [22], an adaptive learning rate optimization algorithm. In TCGA cancer studies, the output of the model is a vector of probabilities converted by the Softmax function, representing the probability to LTS or non-LTS.

#### 2.2.2. Traditional Machine Learning Models

G2PDeep-v2 integrates various traditional machine learning methods, such as Logistic Regression (LR), Support Vector Machine (SVM), Decision Tree (DT), and Random Forest (RF) for providing easy access to the commonly used tools and to serve as benchmark models for comparison with deep-learning approaches. The input for these models is a vector of values concatenated from each type of omics. For logistic regression, it uses an L2 penalty term to deal with multicollinearity problems and penalize insignificant biomarkers. The SVM model uses the radial basis function (RBF) kernel, which makes the data separable using a hyperplane by projecting non-linearly separable data into higher-dimensional space. The decision tree, a nonparametric machine learning algorithm, facilitates training the data without strong assumptions or prior knowledge. The random forest, an ensemble learning method, can handle both linear and non-linear types of data.

### 2.3. Biomarkers Discovery and Annotation

The significant biomarkers associated with phenotypes of interest to researchers are estimated using models in G2PDeep-v2. Based on our previous work [14], the saliency map algorithm is applied to the multi-CNN to identify SNPs highly associated with the phenotype, while the coefficients of traditional ML models are also utilized to pinpoint significant biomarkers. Biomarkers with higher estimated values are considered significant. To facilitate the functional annotation of these identified significant biomarkers, the Gene Set Enrichment Analysis (GSEA) function of GSEApy [23] (version 1.1.11), a Python library, is employed.

### 2.4. Web Server Implementation

G2PDeep-v2 is developed using Model-View-Controller (MVC) architectural pattern and deployed in Docker. This containerized deployment is hosted on a server equipped with an Intel(R) Xeon(R) Gold 6248 CPU and 384 GB of memory, signifying a robust computing environment capable of efficiently handling the computational demands of G2PDeep-v2. G2PDeep-v2 is designed to provide users with a clean and orderly appearance of interface components, reducing the chances of faulty operations and improving user experience. It utilizes high-performance computing resources to guarantee efficient, sustainable, and reliable services with a high volume of tasks. The architectural framework of G2PDeep comprises four modules, complemented by a security policy as illustrated in Figure 2.

#### 2.4.1. Web Interface Module

G2PDeep-v2 provides user-friendly web interface developed using ReactJS [24] and Material UI [25], enterprise-level user interface (UI) libraries. It is designed to be responsive and to render content freely across all screen resolutions on computer and tablet. Plotly [26], a Python graphing library, is used for publication-quality graphs on cross-platform web browsers including Google Chrome, Firefox, Microsoft Edge, and Safari. High-quality interactive charts help users not only summarize the most interesting results easily but also understand the omics-based finding comprehensively.

#### 2.4.2. Core Backend Module

The core backend of G2PDeep-v2 is a middle platform connecting to web interface, database, and the AI platform. It is developed based on the Django REST framework [27], a Python-based powerful and flexible server-side web framework, for managing a high volume of requests and tasks robustly. The Hypertext Transfer Protocol (HTTP) is used to communicate between web interface and backend. The backend integrates different pipelines for dataset creation, models training, and results summarization. It uses Python-based libraries, such as Pandas, NumPy [28] and SciPy [29], to perform a wide variety of mathematical operations on high-dimensional input data and results. The Celery [30], a Python-based extension of Django, schedules model training tasks in a queue and completes expensive operations of training asynchronously.

#### 2.4.3. AI Platform Module

The AI platform is designed for construction, modification, training, and inference of deep learning neural networks and machine learning-based models. The deep learning models and their mathematical optimization are developed based on TensorFlow [31] and Keras [32], high-level deep learning frameworks. The machine learning based models are implemented by scikit-learn [33], free software machine learning library for the Python programming language. Optuna [34], an automatic hyperparameter optimization software framework, provides black box and hyperparameter optimization to maximize the performance of the deep learning and machine learning models.

#### 2.4.4. Database Module

MySQL [35] and Redis [36] databases are used in G2PDeep-v2. MySQL, a relational database, enables meaningful information by joining various organized tables. It manages various multi-omics data, project information, modeling information, training information, and user information. Redis is a NoSQL database and in-memory database, extremely fast in reading and writing the data in random access memory. Redis stores the model training information and details of scheduler, bring the reliability of data storage and transactions during multiple tasks processing.

#### 2.4.5. Security Policy

The G2PDeep-v2 leverages JSON Web Token (JWT) token [37] to control the access to private datasets and models. The JWT is a protocol providing authentication, authorization, and other security features for enterprise applications. Users can create an account by filling out a registration form on the sign-up page with the required information. The activation link for the new account is then sent to users. Users can log into G2PDeep using their registered username and password. The login credential remains valid for 12 h, providing access without having to prompt the user to log in again.

## 3. Results

### 3.1. Overview of the Web Server

The overview of G2PDeep-v2 is depicted in Figure 3. Starting from a multi-omics dataset, G2PDeep-v2 integrates samples from each type of multi-omics and splits merged samples into five equally sized sets with 5-fold cross-validation. G2PDeep-v2 provides a variety of machine learning and deep learning models, including our proposed multi-CNN, Logistic Regression (LR), Support Vector Machine (SVM), Decision Tree (DT), and Random Forest (RF). The platform also features a web-based interactive interface that allows users to create, train, and monitor the performance of these models, which is a unique aspect of bioinformatics. All models are trained using our high-performance computing resources and stored in the database for future inference. G2PDeep-v2 provides prediction for large-scale datasets, and visualization for predicted results and biomarkers associated with corresponding phenotypes. The results of Gene Set Enrichment Analysis (GSEA) for these biomarkers are generated automatically. It also provides complete documentation on the website, including a user guide describing all tools, examples, and frequently asked questions. To accelerate scientific research for survival analysis in cancer studies, we utilized G2PDeep-v2 and established biomarkers associated with long-term survival for 23 cancer studies.

#### 3.1.1. Dataset Creation

To initiate the use of G2PDeep-v2, the pivotal first step involves creating datasets. G2PDeep-v2 allows users to create datasets with two options: uploading a CSV file or transferring data from a link (see Figure 4A). For a small dataset (up to 50 MB), users can create a dataset by uploading their own data from their local machine. For a large dataset (up to 10 GB), users can enter a shared link of data from Google Drive, OneDrive, CyVerse Data Store [38,39], or other public repositories. Users can upload multi-omics data, including gene expression, miRNA expression, DNA methylation, protein expression SNP, and CNV. Once the files are uploaded, G2PDeep-v2 performs *z*-score normalization for each expression sample and imputes missing values automatically. To merge multi-omics data from various sources, the datasets must share a column with unique IDs for each sample. By combining data from multiple sources, users can create more comprehensive datasets that may be better suited to their research questions. Users can also enter the type of data source to indicate whether the dataset is from humans or plants. The G2PDeep-v2 validates uploaded files to guarantee the data can be used in model creation. For any invalid format or unsupported data type, it has a function to stop data creation and notify users about the corresponding error message. It also shows a progress bar with duration remaining, allowing users to monitor the status of the dataset creation. The created datasets are private and only retrievable by the owners of the datasets. G2PDeep-v2 supports user’s needs of sharing data with the community after anonymization by removing identifiable information for samples, making it available to other researchers to work on same data and share insights while protecting dataset privacy. G2PDeep-v2 also integrates the publicly available datasets, such as 23 TCGA cancer datasets, SoyNAM datasets [40] and Bandillo’s SNP datasets [41] (see Figure 4B). Comprehensive details for each dataset, including links to data, type of data, number of samples, and features, are directly retrievable from the website. Once the datasets are created, users can build their models for the datasets.

#### 3.1.2. Model Creation

Transitioning to model creation, G2PDeep-v2 emphasizes customization as a key feature. Hyperparameters, critical components influencing machine learning model performance, can be tailored by users on the Model Creation page (See Figure 5). The range of suggested hyperparameters and training parameters for models in G2PDeep-v2 are shown in Appendix A. Users can also select up to three different types of data as input and determine whether the model is designed for quantitative phenotype prediction or categorical phenotype prediction.

To strike a balance between training speed and model performance, G2PDeep-v2 provides three strategic options for setting hyperparameters. The first involves using default pre-tuned hyperparameters based on models created using data from 23 different TCGA studies and WGRS dataset for SCN resistance, enabling users to quickly generate models without additional tuning. Alternatively, users can opt for the second strategy, customizing hyperparameters through an interactive interface, aligning their models with specific datasets and research questions. The third strategy employs an automated hyperparameter search using a Bayesian optimization algorithm [42], efficiently exploring a large search space to identify optimal hyperparameters challenging to pinpoint through manual tuning.

Once users complete model creation, G2PDeep-v2 automatically saves the model as a private entry in the database. Users can conveniently access and manage their private and public models, along with corresponding configurations. Additionally, the platform supports model sharing within the community, fostering collaboration and knowledge exchange.

#### 3.1.3. Project for Model Training and Evaluation

Once the dataset and model are prepared, users can seamlessly leverage G2PDeep-v2 to train models using the uploaded datasets. On the Project Creation page, users can conveniently access all publicly available models as well as their private models, categorized based on the type of multi-omics data they are interested in. To initiate a new project of models training, users are prompted to select a dataset for each type of multi-omics data to serve as input for the model. After dataset selection, users have the flexibility to experiment with different hyperparameter-setting strategies to identify the optimal configuration for their specific data. Upon submission of the project, it enters a task queue, awaiting allocation of computing resources. The project settings and model configurations are securely stored in the database. Notably, for cancer data, the server typically takes around 2 h to train a model using automated hyperparameter tuning settings, involving 400 training samples across three types of multi-omics data and only CPU resources.

Users can track progress via a detailed summary page throughout the model training process. A progress bar with duration and percentage is displayed on the summary page, along with the estimated time to completion and model information. Further insights into the model, dataset, and training information are accessible on the Detail page, as illustrated in Figure 6. Dataset details include names, omics types, number of samples, and features, presented in a clear tabular format. Model information encompasses the model type and a diagram illustrating the kernel size and number of filters for each layer. The learning curve graphically portrays the performance of model on both training and validation datasets, aiding in assessing overfitting or underfitting.

Once the model reaches optimal training, G2PDeep-v2 provides interactive plots illustrating predicted results and model performance on both training and validation datasets. For categorical phenotype prediction tasks, a bar chart depicts the frequency of predicted labels alongside ground truth. Receiver Operating Characteristic (ROC) curves and Precision-Recall curves offer a visual representation of the diagnostic capabilities of the model. In cases of quantitative phenotype prediction tasks, a scatter plot compares predicted values with ground truth, accompanied by metrics like the Pearson correlation coefficient (PCC) and coefficient of determination (R squared). All predicted results and interactive plots are downloadable as CSV files and PNG images.

#### 3.1.4. Prediction and Significant Biomarkers Discovery

Users can utilize G2PDeep-v2 to make predictions and visualize results using multi-omics data and a well-trained model. The predictions take, on average, less than 30 s to predict phenotype and marker significance for 1000 samples. Precisely, users can effortlessly input data by uploading a CSV file directly to the server for each type of multi-omics data. The system performs thorough validation, ensuring adherence to the required format, and promptly notifies users of any invalid input data through error notification. Notably, the system accommodates up to 10,000 samples, and a user-friendly progress bar allows for real-time monitoring of prediction status. All predicted results are securely stored in the database, readily retrievable for future analysis and comparison.

Upon completion, G2PDeep-v2 generates a bar chart illustrating predicted values and a plot highlighting significant biomarkers (shown in Figure 7A). Users retain the flexibility to adjust the number of displayed biomarkers by setting a threshold based on the highest saliency values, focusing on the most relevant biomarkers for their specific research requirements. The plot presents significant biomarkers sorted by decreasing saliency values, and this information can be conveniently saved as a CSV file. G2PDeep-v2 also provides GSEA for significant biomarkers. It performs GSEA based on KEGG [16] and Reactome [17] pathway databases (shown in Figure 7B), which are widely used and comprehensive resources for pathway information. In cases where the biomarkers are not genes, such as CpG islands identified from methylation data, G2PDeep-v2 converts these markers to the corresponding neighboring gene that they regulate to fetch significance. It also provides users with a scatterplot for top 10 enriched pathways from KEGG and Reactome for the gene sets, making it easy to gain insights into the molecular mechanisms underlying complex diseases and other biological phenomena. Detailed information on enriched pathways is presented in tabular form, including corresponding *p*-values, adjusted *p*-values, and gene sets. Additionally, a table listing literature evidence associated with significant biomarkers and relevant cancer or other studies enhances the interpretability of the results.

#### 3.1.5. Study Results in G2PDeep-v2

We regularly update and share the outcomes of cancer studies on the Study Results Page within G2PDeep-v2. Users can effortlessly access and retrieve results tailored to their specific interests, thereby facilitating enhanced accessibility for subsequent analysis and exploration.

Currently in G2PDeep-v2, we conducted several comprehensive studies using the 23 TCGA cancer studies dataset encompassing six distinct types of multi-omics data independently. The diverse array of multi-omics data, including gene expression, miRNA expression, DNA methylation, protein expression SNP, and CNV, was downloaded from the Broad Institute Fire Browse portal [43]. To ensure a robust analysis, we systematically created 41 datasets for each cancer study. These datasets include individual types of omics (6 datasets), combinations of two omics (15 datasets), and combinations of three omics (20 datasets). The phenotypes of these studies are long-term survival (LTS) and non-long-term survival (non-LTS) groups. The LTS is defined as survival > 3 years after diagnosis, and the non-LTS is defined as survival ≤ 3 years. Individuals who survived with the last follow-up of ≤3 years are excluded from further analysis.

To make 23 TCGA studies applicable to both ideal scenarios and real-world conditions, we categorized them into two types: studies with uniform multi-omics data and those with non-uniform multi-omics data. In the context of ideal scenarios, uniform data denotes that patient cohorts in these studies encompass all six types of multi-omics data, while non-uniform data for real-world conditions indicates that cohorts may lack some types of multi-omics data. Precisely, the uniform data can be considered a subset of the non-uniform data. The studies with uniform omics data are tailored to investigate the significance of multi-omics data combinations. Due to limitations in the cohort of patients, we specifically designated six out of the total 23 studies as studies with uniform omics data. On the other hand, studies with non-uniform data are designed to explore biomarkers under scenarios that more closely mirror the complexities of real-world conditions. We finally made a total of 23 studies specifically with non-uniform data. The specifics of uniform and non-uniform multi-omics data for each cancer study, including information such as sequencing platforms, the number of features, and samples, are comprehensively listed in Table 2 and Table 3, respectively.

The G2PDeep-v2 conducted a thorough analysis of phenotype prediction using both studies with uniform and non-uniform multi-omics data. Various models, including our proposed multi-CNN, LR [44], SVM [45], DT [22,46], and RF [47], were employed for predictions. To ensure reproducibility, the data for each cancer study underwent a systematic division into a training dataset (60% of the entire data) for model training, a validation dataset (20% of the entire data) for hyper-parameter tuning, and a test dataset (20% of the entire data) to evaluate model performance. The model was constructed in each cross-validation iteration and rigorously evaluated on the designated test set. Quantification of predictive performance was achieved by calculating the mean area under the curve (AUC) over a 5-fold cross-validation framework. Figure 8 illustrates that G2PDeep-v2 using our proposed multi-CNN outperforms other ML models in predicting phenotypes for the Skin Cutaneous Melanoma (SKCM) study with uniform multi-omics data. Based on the metrics recorded for models applied to both studies with uniform and non-uniform multi-omics, as depicted in Appendix A, respectively, G2PDeep-v2 using our proposed multi-CNN also outperforms or competes effectively with other ML models across most of the cancer studies. All performance details are conveniently accessible on the Study Result Page, providing a consolidated view of the effectiveness of models across various multi-omics data scenarios for user convenience. Furthermore, we expanded upon the study results by incorporating significant biomarkers and conducting corresponding GSEA.

### 3.2. Application of G2PDeep-v2

#### 3.2.1. Use Case #1: Long-Term-Survival Prediction and Markers Discovery for Cancer

The motivation for this use case is to highlight the advantages of G2PDeep-v2 for long-term survival prediction and biomarker discovery in Breast Invasive Carcinoma (BRCA) cancer. We used G2PDeep-v2 to predict the phenotype of BRCA patients based on their multi-omics data, including gene expression, miRNA expression, DNA methylation, protein expression, SNP, and CNV data. We created and trained deep learning models to accurately predict the long-term survival of BRCA patients. According to our results (See Appendix A), the best model trained on three combinations of omics is the CNN model, which achieved a mean AUC score of 0.907. The three combinations of omics are gene expression, miRNA expression, and SNP. We generated significant biomarkers and sorted them by saliency values. We selected the biomarkers with the top 100 highest saliency values and compared these biomarkers with oncogenes from the OncoKB database [48]. We found that 6 out of the 100 genes are oncogenes (see Appendix A). We then performed GSEA on these 100 genes and found seven pathways with *p*-values lower than 0.05. We noticed that most of the enriched pathways are related to breast cancer development (see Appendix A). Todd et al. [49] have reported that breast cancer with aberrant activation of the PI3K pathway can be identified by somatic mutations, suggesting potential dependence on the phosphatidylinositol signaling system pathway. Klara et al. [50] reported that N-glycosylation of breast cancer cells during metastasis is observed in a site-specific manner, highlighting the significance of high-mannose, fucosylated, and complex N-glycans as potential diagnostic markers and therapeutic targets in metastatic breast cancer. The Notch signaling pathway promotes tumor progression and survival and induces a breast cancer stem cell (CSC) phenotype [51]. This evidence supports the relevance of the identified biomarkers and their contribution towards these predictions.

#### 3.2.2. Use Case #2: Disease Resistance Prediction for Soybean Cyst Nematode (SCN) in Soybean 1066 Lines

In this use case, we tested G2PDeep-v2 for Soybean Cyst Nematode (SCN) using Copy Number Variation (CNV) data, extracted from publicly available Whole-Genome Resequencing (WGRS) datasets for 1066 Soybean accessions [52]. The dataset consisted of multiple phenotypes, 228 samples from this dataset had readings for SCN phenotype, with class categories, Susceptible (S) and Resistant (R). G2PDeep’s multi-CNN model was trained on 80 percent of this dataset, and its performance was evaluated on 5-fold cross-validation, using the AUC. The model performed consistently well on all 5-folds. To interpret the model’s predictions and identify main genomic regions responsible for prediction of SCN resistance, we implemented a saliency map approach. This approach ranked the resultant SNP list based on the saliency values. In a further step to simplify rankings, saliency value was converted to dense rank; the higher the saliency value, lower it is rank would be. SNPs are mapped to genes based on their chromosomal positions using the soybean genome to generate a corresponding gene list. Based on the ranked gene list, the model identified a novel gene *Glyma.13g030200* (as shown in Appendix A), which ranked tenth in the saliency list. Interestingly, protein from the same family was previously published as a candidate for nematode resistance in rice [53]. To validate these results further, we looked at the regulatory aspects, to explore the Transcription Factors (TF) binding to *Glyma.13g030200* promoter region. GenVarX tool [54] in SoyKB [52,55], identified 81 TF binding sites within a 2 kb upstream region of the new candidate gene. Notably, 37 of these sites were particularly found to contain variantsTo explore Indels in the identified promotor regions, SNPViz tool [56,57,58] in SoyKB was utilized. This identified large insertions within the promotor region (as shown in Appendix A), which can potentially regulate the function of this gene affecting its role in SCN resistance. Further functional enrichment was performed on the resulting gene list, using GProfiler [59], to analyze Gene Ontology (GO) and KEGG pathway enrichment, where results revealed GO terms associated with defense response and stress response (as shown in Appendix A). The overall findings suggest *Glyma.13g030200* as a promising candidate which can be further investigated for SCN resistance phenotype. Further studies may be required to experimentally validate its precise function in SCN resistance.

## 4. Discussion

G2PDeep-v2 webserver is developed as a one-stop shop platform that addresses the need for efficient and accurate phenotype predictions from multi-omics data with customizable deep learning and machine learning models for any organisms. G2PDeep-v2 is the first web server that allows models to be created, trained with automated hyperparameter tuning, and used for inference on multi-omics data uploaded by researchers. We have deployed G2PDeep-v2 on a server equipped with both CPU and GPU resources to expedite model training and inference processes. Performance, compatibility, usability, and interpretability are all central principles of G2PDeep-v2. G2PDeep-v2 integrates numerous deep learning and machine learning models that are well-trained on 23 different TCGA cancer studies, SoyNAM, and Bandillo’s SNP datasets, allowing researchers to reuse these models to predict phenotypes and identify significant biomarkers for biomedical and agribiotech purposes. In the context of G2PDeep-v2, significant refers to features identified by the CNN as important contributors to model predictions, rather than statistically significant features in the classical sense, which limits the interpretability of the results in traditional statistical terms. It has applications for predicting phenotypes in a wide range of research domains, including human and agriculture. It can also further help uncover the specific multi-omics data types that may be best suited for respective phenotype predictions.

In many real-world scenarios, such as medical research and rare disease studies, obtaining sufficient labeled data remains a major challenge. To address this issue, we plan to incorporate meta-learning techniques that enable models to learn effectively from limited data by leveraging prior knowledge from related tasks or experiences. To mitigate batch effects in multi-omics datasets, we also plan to employ contrastive learning to derive feature representations that are invariant to batch effects and robust to missing values. By comparing representations from different batches, the model can identify shared biological patterns independent of technical variations. Currently, G2PDeep-V2 supports a maximum combination of three omics data types, as samples containing four or more omics types are extremely limited in most available datasets. We aim to expand the framework to accommodate higher-order omics integration as more comprehensive datasets become available. Furthermore, we plan to enhance G2PDeep-V2 to support multi-class prediction tasks. As the current version does not yet include cross-species analysis, we plan to implement a new cross-organism validation module and perform comprehensive evaluations to rigorously assess its performance. We also plan to incorporate advanced deep learning interpretability methods, such as Grad-CAM, Layer-wise Relevance Propagation (LRP), and SHAP-based analyses. In future versions, we plan to integrate more sophisticated imputation techniques, such as KNN or mean imputation, to further enhance flexibility and performance. Additionally, future versions will include survival models and classical evaluation metrics, such as the concordance index (C-index) and Kaplan–Meier curves, as well as functionality to handle data heterogeneity, skewed distributions, and outliers in phenotypic data with the idea from Wu et al. [60]. Currently, we are working on combining scRNA-seq with bulk RNA-seq to improve the accuracy and resolution of transcriptomic analysis. By integrating scRNA-seq and bulk RNA-seq data, we can identify cell-type-specific gene expression patterns in complex tissues, enabling a deeper understanding of cellular heterogeneity and the identification of new biomarkers, than can be achieved by bulk transcriptomics alone. G2PDeep-v2 features will continue to expand and develop in response to the evolving needs of the research community.

## 5. Conclusions

G2PDeep-v2 is a novel and comprehensive web-platform that enables researchers to perform phenotype prediction, biomarker discovery, and GSEA for a range of applications in research in human disease and plant breeding. G2PDeep-v2 allows for easy customization and optimization of models without the need for extensive experience in machine learning. By integrating various multi-omics datasets and pre-trained models, G2PDeep-v2 enables the creation of robust and reproducible predictions and biomarkers, while also providing access to a wealth of downstream analysis tools and results from multiple studies. Overall, G2PDeep-v2 represents a single one-stop-shop solution for phenotype predictions, with potential applications in precision medicine, drug discovery, precision agriculture, genomic epidemiology and other areas of research that rely on complex omics data.

## Figures and Tables

**Figure 1 biomolecules-15-01673-f001:**
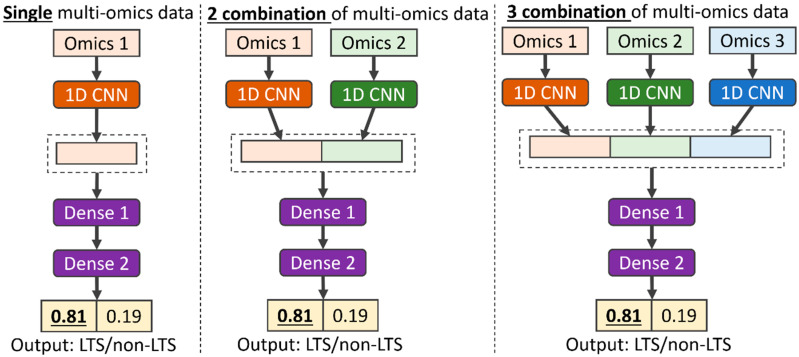
An example architecture of the multi-CNN model designed for long-term survival prediction using input data with single, two combinations, and three combinations of multi-omics data.

**Figure 2 biomolecules-15-01673-f002:**
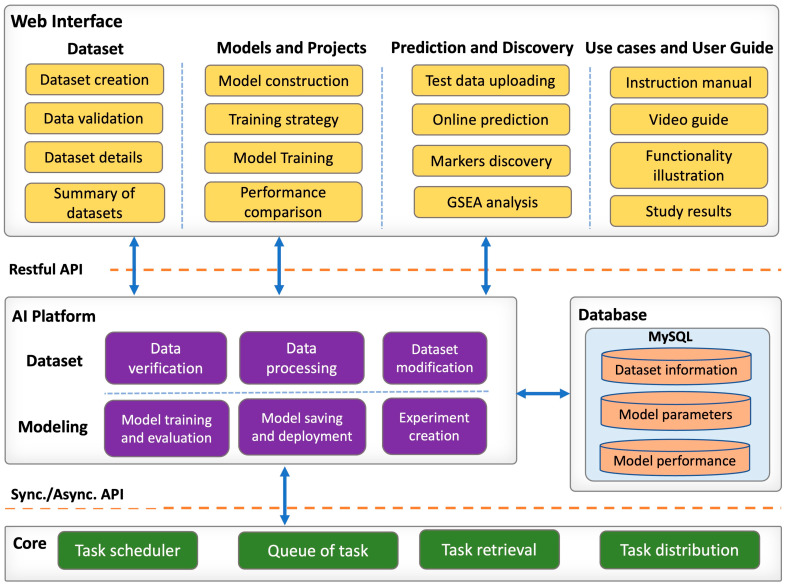
The architecture of G2PDeep. The architecture consists of four modules, and these modules communicate with each other via appropriate APIs, as indicated by the blue arrows.

**Figure 3 biomolecules-15-01673-f003:**
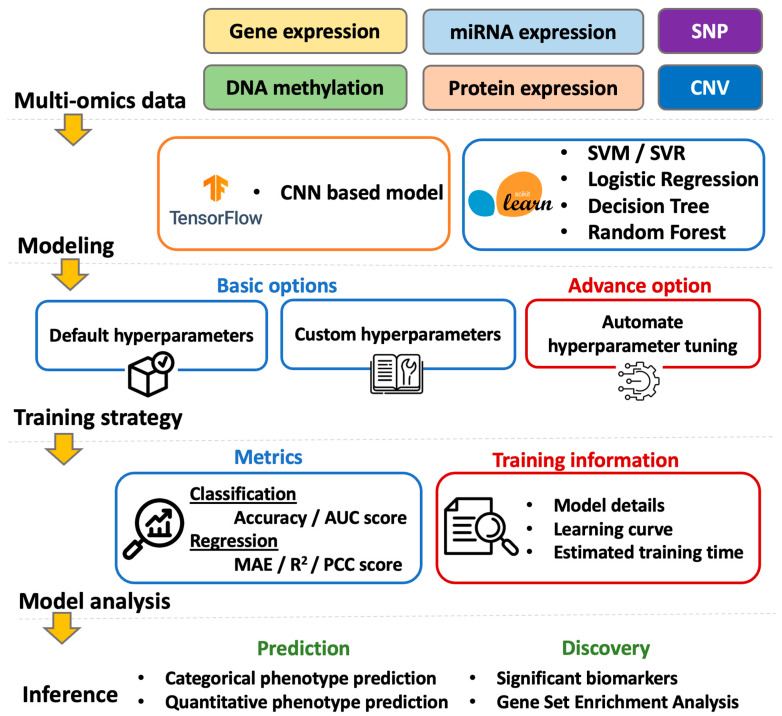
Overview of G2PDeep-v2.

**Figure 4 biomolecules-15-01673-f004:**
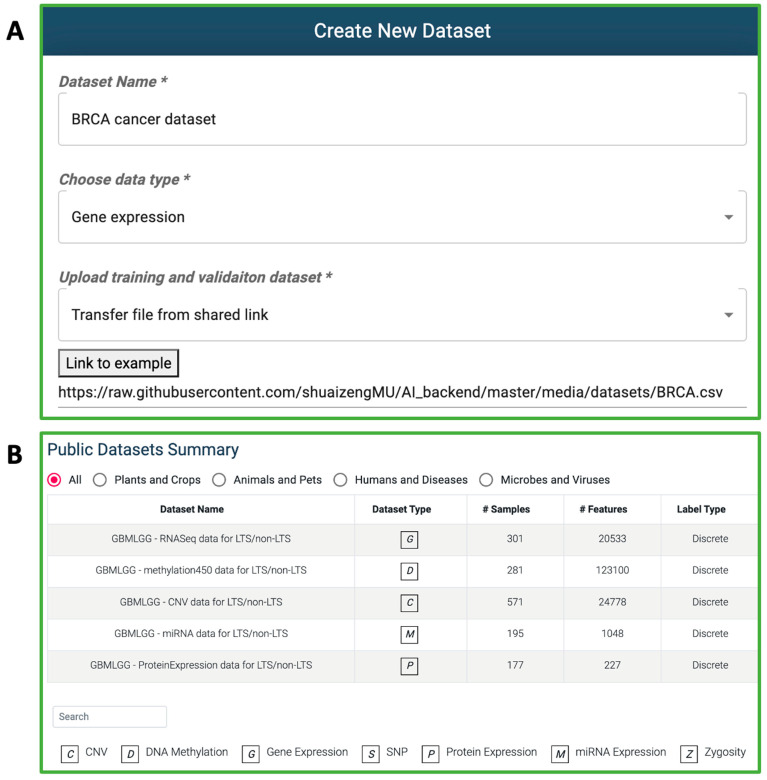
Dataset creation and retrieval in G2PDeep-v2. (**A**) Example of dataset creation by a shared link to the required data, indicated with a * symbol. (**B**) Publicly available datasets are shown with structured information.

**Figure 5 biomolecules-15-01673-f005:**
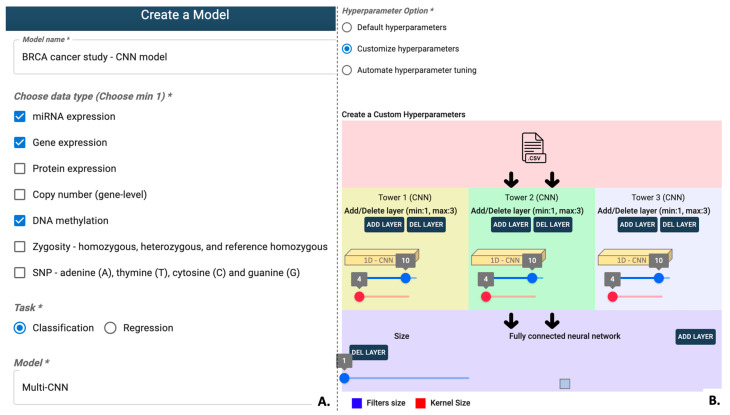
Interactive chart to configure the deep-learning model in G2PDeep-v2. (**A**) Required options (indicated with a * symbol) for inputting details such as the model, task, and input data. (**B**) Hyperparameters tuning options.

**Figure 6 biomolecules-15-01673-f006:**
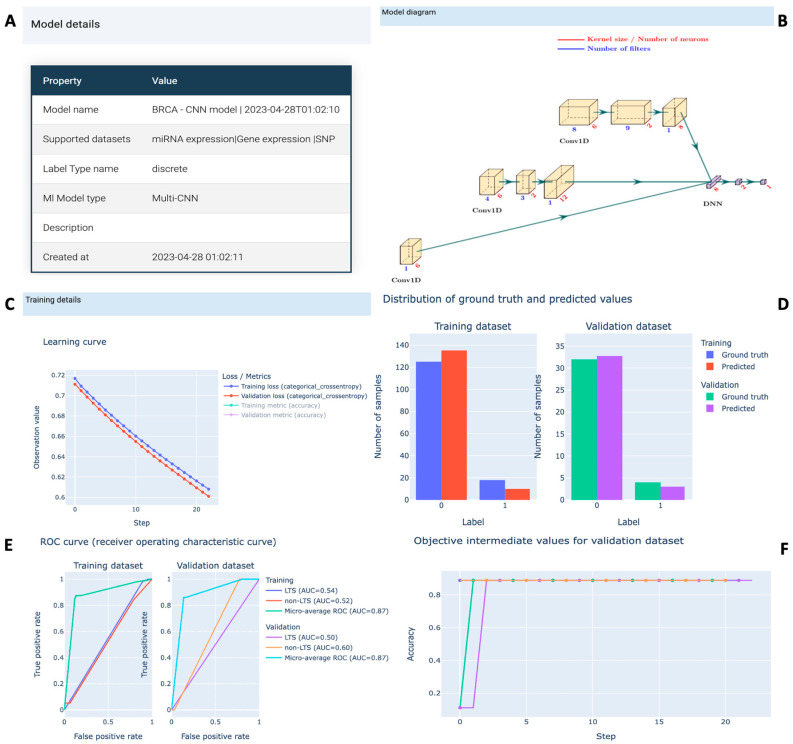
Project page in G2PDeep-v2. (**A**) Model summary showing the type of model and corresponding training dataset; (**B**) visualization of the multi-CNN architecture, illustrating the convolutional and fully connected layers used for multi-omics feature extraction and integration; (**C**) learning curves showing training and validation loss across epochs, allowing users to monitor model convergence and potential overfitting; (**D**) distribution of ground and predicted values for training and validation datasets; (**E**) ROC for phenotype prediction, demonstrating the discriminative ability of model on training and validation datasets; (**F**) optimization history from the Bayesian hyperparameter tuning process, highlighting how model performance improves over successive iterations of parameter adjustment. Each line represents a single trial.

**Figure 7 biomolecules-15-01673-f007:**
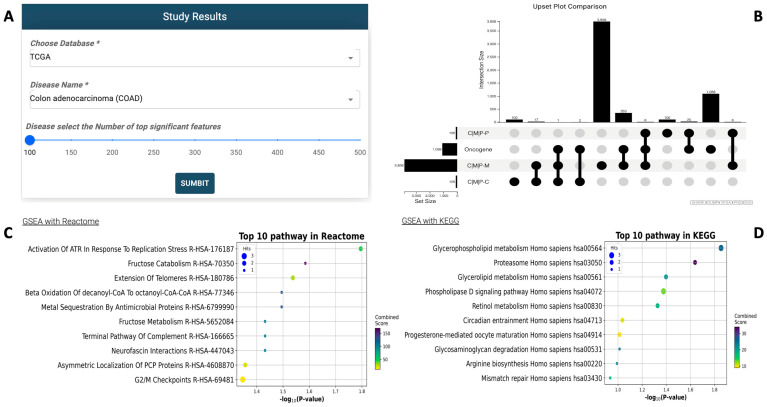
Study Results page in G2PDeep-v2. (**A**) Panel to select study with required options, as indicated by a * symbol; (**B**) Upset plot shows overlapping significant biomarkers; (**C**) GSEA with Reactome for significant biomarkers; (**D**) GSEA with KEGG for significant biomarkers.

**Figure 8 biomolecules-15-01673-f008:**
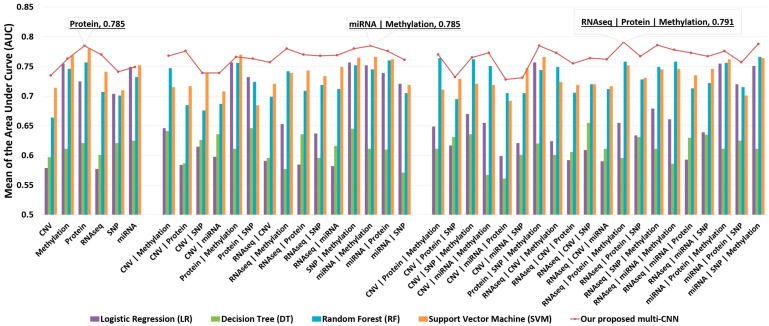
Comparison of model performance on 41 datasets from the Skin Cutaneous Melanoma study. Each model was trained and evaluated separately on individual datasets, and the mean area under the AUC was calculated. The results show that the proposed multi-CNN model (dotted line) consistently outperforms traditional machine learning methods (bars), including logistic regression, support vector machines, decision trees, and random forests.

**Table 1 biomolecules-15-01673-t001:** Comparison of functionalities between the previous and latest versions of G2PDeep.

Categories	Functionality	G2PDeep-v1	G2PDeep-v2
Dataset creation	single nucleotide polymorphisms (SNP)/Zygosity	✔	✔
	gene expression		✔
	copy number variation (CNV)		✔
	Protein expression		✔
	microRNA (miRNA) expression		✔
	DNA Methylation		✔
Custom models	dual-CNN/multi-CNN	✔	✔
	Support Vector Machine (SVM)		✔
	Logistic Regression (LR)		✔
	Random Forest (RF)		✔
	Decision Tree (DT)		✔
	Multiple inputs		✔
Task	Regression	✔	✔
	Classification		✔
Model training	Online training	✔	✔
	Training monitoring	✔	✔
	Automate hyperparameter tunning		✔
	Hyperparameter tunning monitoring		✔
Online prediction	Prediction with test dataset	✔	✔
Marker discovery	Identifying significant markers	✔	✔
	GSEA with KEGG/Reactome		✔
	Studies related to significant markers		✔

**Table 2 biomolecules-15-01673-t002:** Uniform dataset for 6 different TCGA cancer studies.

Study	Number of Samples (LTS/Non-LTS)	Number of Features					
		Gene Expression	miRNA Expression	DNA Methylation	Protein Expression	SNP	CNV
BLCA	42 (15/27)	20,533	1048	300,869	225	18,634	24,778
HNSC	39 (14/25)	20,533	1048	300,973	239	17,796	24,778
LUAD	33 (16/17)	20,533	1048	300,822	239	18,950	24,778
LUSC	28 (15/13)	20,533	1048	300,970	239	18,822	24,778
SARC	26 (15/11)	20,533	1048	299,776	219	12,422	24,778
SKCM	41 (29/12)	20,533	1048	300,455	225	19,488	24,778

**Table 3 biomolecules-15-01673-t003:** Dataset for 23 different TCGA cancer studies.

Study	Number of Samples (LTS/Non-LTS)				
	Gene Expression	miRNA Expression	DNA Methylation	Protein Expression	SNP
ACC	62 (44/18)	63 (44/19)	63 (44/19)	36 (28/8)	73 (50/23)
BLCA	248 (87/161)	250 (89/161)	252 (89/163)	215 (76/139)	252 (89/163)
BRCA	506 (437/69)	344 (296/48)	364 (314/50)	410 (351/59)	455 (395/60)
CESC	146 (91/55)	146 (91/55)	146 (91/55)	65 (44/21)	138 (86/52)
CHOL	26 (11/15)	26 (11/15)	26 (11/15)	22 (9/13)	26 (11/15)
COAD	126 (78/48)	91 (56/35)	130 (81/49)	133 (79/54)	172 (101/71)
ESCA	86 (17/69)	87 (18/69)	87 (18/69)	51 (12/39)	86 (18/68)
HNSC	327 (144/183)	298 (128/170)	331 (145/186)	230 (89/141)	318 (135/183)
KICH	53 (47/6)	53 (47/6)	53 (47/6)	51 (45/6)	53 (47/6)
KIRC	404 (293/111)	177 (132/45)	228 (157/71)	246 (173/73)	226 (173/53)
KIRP	127 (100/27)	127 (100/27)	120 (94/26)	95 (75/20)	120 (94/26)
LIHC	195 (91/104)	195 (92/103)	199 (94/105)	109 (35/74)	189 (89/100)
LUAD	270 (133/137)	223 (109/114)	230 (112/118)	204 (102/102)	269 (133/136)
LUSC	305 (149/156)	196 (95/101)	222 (111/111)	204 (106/98)	302 (146/156)
MESO	80 (14/66)	80 (14/66)	80 (14/66)	58 (8/50)	76 (14/62)
PAAD	108 (20/88)	108 (20/88)	114 (21/93)	70 (11/59)	112 (21/91)
READ	38 (27/11)	33 (23/10)	40 (29/11)	46 (30/16)	49 (36/13)
SARC	177 (108/69)	177 (108/69)	179 (109/70)	150 (87/63)	159 (96/63)
SKCM	335 (227/108)	322 (219/103)	336 (227/109)	236 (152/84)	334 (226/108)
STAD	196 (48/148)	184 (47/137)	189 (49/140)	170 (38/132)	208 (49/159)
THCA	208 (199/9)	209 (200/9)	210 (201/9)	169 (160/9)	205 (198/7)
UCEC	69 (44/25)	183 (127/56)	193 (137/56)	217 (163/54)	273 (208/65)
UCS	42 (12/30)	41 (12/29)	42 (12/30)	36 (8/28)	42 (12/30)

## Data Availability

The G2PDeep-v2 server is publicly available at https://g2pdeep.org (accessed on 1 August 2023).

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
