# Peer review of "G2PDeep-v2: A Web-Based Deep-Learning Framework for Phenotype Prediction and Biomarker Discovery for All Organisms Using Multi-Omics Data"

_biomolecules, 2025, doi:10.3390/biom15121673_

Round 1
Reviewer 1 Report (Previous Reviewer 1)
Comments and Suggestions for Authors
The authors have adequately addressed all previously raised concerns
Author Response
Comment: The authors have adequately addressed all previously raised concerns
Response: We thank the reviewers for their positive evaluation and appreciation of our revisions.
Reviewer 2 Report (Previous Reviewer 2)
Comments and Suggestions for Authors
No further comments.
Author Response
Comment: No further comments.
Response: We appreciate the reviewer’s time and consideration.
Reviewer 3 Report (Previous Reviewer 3)
Comments and Suggestions for Authors
I confirm the previous judgement. I do not see any substantial change in the manuscript that would correct its previous approach and steer it in the right direction. The manuscript does not fall into the area of interest of Biomolecules: "The journal has a strong focus on structures and functions of bioactive and biogenic substances, molecular mechanisms with biological and medical implications as well as biomaterials and their applications."
In other words: this manuscript is not focused on the same topic of the journal. The article is focused on the description of the bioinformatics tool. It is evident starting from the title, and also the abstract, where I do not find reference to structures and functions of bioactive and biogenic substances or molecular mechanisms investigated into the manuscript (and this is the natural consequence of the fact that the article is focused on the software development, not on the interests of the journal). And let it be clear that it is not enough to change the title or add a sentence to the abstract. The article does not focus on identifying biomarkers associated with patient outcomes in cancer and disease resistance in soybean (a combination of topics suitable for demonstrating the versatility of the software), as the journal's aims would require, and as the authors seem to want to claim in their response.
The authors believe that, since their software was developed for applications in the journal's field of interest, it is possible to publish in this journal a manuscript describing the software. I believe that this is not the case; quite simply, the article is more suited to a journal that focuses on software developed for biomedical applications, and must be evaluated in the context of that type of journals.
Author Response
Comment: For Authors I confirm the previous judgement. I do not see any substantial change in the manuscript that would correct its previous approach and steer it in the right direction. The manuscript does not fall into the area of interest of Biomolecules: "The journal has a strong focus on structures and functions of bioactive and biogenic substances, molecular mechanisms with biological and medical implications as well as biomaterials and their applications." In other words: this manuscript is not focused on the same topic of the journal. The article is focused on the description of the bioinformatics tool. It is evident starting from the title, and also the abstract, where I do not find reference to structures and functions of bioactive and biogenic substances or molecular mechanisms investigated into the manuscript (and this is the natural consequence of the fact that the article is focused on the software development, not on the interests of the journal). And let it be clear that it is not enough to change the title or add a sentence to the abstract. The article does not focus on identifying biomarkers associated with patient outcomes in cancer and disease resistance in soybean (a combination of topics suitable for demonstrating the versatility of the software), as the journal's aims would require, and as the authors seem to want to claim in their response. The authors believe that, since their software was developed for applications in the journal's field of interest, it is possible to publish in this journal a manuscript describing the software. I believe that this is not the case; quite simply, the article is more suited to a journal that focuses on software developed for biomedical applications, and must be evaluated in the context of that type of journals.
Response: We sincerely thank reviewer for the detailed comments and for carefully evaluating our revised manuscript. We respectfully disagree with the assessment that our work falls outside the scope of Biomolecules. While our study presents a bioinformatics platform, its focus and applications are directly aligned with the journal’s emphasis on elucidating molecular mechanisms with biological and medical implications. Specifically, G2PDeep-v2 is designed for multi-omics data integration and biomarker discovery—both central to biomolecular research. The platform enables meaningful biological and medical insights, such as identifying biomarkers associated with patient outcomes in cancer and disease resistance in soybean. We therefore believe that the manuscript is well within the scope of Biomolecules and represents a valuable methodological advancement that supports biomolecular and biomedical research. We respectfully hope that the Editor and Reviewer will agree that this work makes a suitable and meaningful contribution to the journal.
Reviewer 4 Report (Previous Reviewer 4)
Comments and Suggestions for Authors
The authors have addressed my comments included in the improved version.
Author Response
Comment: The authors have addressed my comments included in the improved version.
Response: We sincerely thank the reviewer for acknowledging our revisions and for their positive evaluation.
Reviewer 5 Report (New Reviewer)
Comments and Suggestions for Authors
The manuscript introduces G2PDeep-V2, a powerful and comprehensively designed web platform dedicated to addressing the challenges of multi-omics data integration, phenotype prediction, and biomarker discovery. The platform's main value lies in offering a "one-stop," end-to-end deep-learning framework that integrates various models (including Multi-CNN), automated hyperparameter tuning based on Bayesian optimization, and complete biomarker functional annotation (GSEA). This significantly lowers the barrier for non-bioinformatician users to utilize complex deep-learning techniques. However, the manuscript requires further refinement and clarification regarding some key technical details and the presentation of core results.
1. G2PDeep-V2 supports six types of omics data for preprocessing, yet the Multi-CNN model architecture and platform description suggest it only supports a maximum combination of three omics data types as input for prediction. What is the theoretical or technical justification for restricting the input to three omics data types? If a user possesses four or more omics data types, what specific strategy does the platform recommend for effective analysis?
2. Missing Value Imputation Justification: In the data preprocessing step, missing values for expression data (e.g., gene, miRNA, DNA methylation, protein) are simply set to 0 (zero-imputation), while SNP and CNV data are left unimputed. Have the authors evaluated the potential bias introduced by this simplistic zero-imputation method on model performance and subsequent biomarker identification? Please provide the rationale for choosing zero-imputation over more sophisticated imputation techniques (e.g., KNN or mean imputation).
3. Biomarker Interpretation Robustness: The manuscript states that the Multi-CNN model uses the Saliency Map algorithm to identify important biomarkers. Given the known sensitivity of Saliency Maps to small input perturbations, have the authors attempted or considered other more robust or advanced deep-learning interpretability methods (such as Grad-CAM, LRP, or SHAP-based analysis)? A justification for using Saliency Map as the primary method for multi-omics biomarker discovery should be provided.
4. Performance Metrics and Computational Resources: The manuscript mentions that training a typical model with automated hyperparameter tuning takes about 2 hours and only utilizes CPU resources. Given the general computational demands of deep learning, does G2PDeep-V2 support the use of GPUs for model training to accelerate tasks? Furthermore, the Results section currently lacks quantitative performance evaluations (e.g., AUC, R-squared, or PCC) for the 23 TCGA cancer studies and the SCN case study. Please supplement the manuscript with a table clearly presenting the key prediction performance metrics for the model in these case studies under different omics combinations.
Author Response
Comment 1: G2PDeep-V2 supports six types of omics data for preprocessing, yet the Multi-CNN model architecture and platform description suggest it only supports a maximum combination of three omics data types as input for prediction. What is the theoretical or technical justification for restricting the input to three omics data types? If a user possesses four or more omics data types, what specific strategy does the platform recommend for effective analysis?
Response 1: We appreciate the reviewer’s insightful question. In real-world datasets, samples containing more than three omics data types are extremely limited. For example, in our TCGA case studies, only a small subset of patients have three omics types available, while those with four or more omics types are too few to train a reliable model. Therefore, G2PDeep-V2 currently supports a maximum combination of three omics data types to ensure sufficient sample size and stable model training. We have also mentioned this limitation in the Discussion section, as shown below.
“Currently, G2PDeep-V2 supports a maximum combination of three omics data types, as samples containing four or more omics types are extremely limited in most available datasets. We aim to expand the framework to accommodate higher-order omics integration as more comprehensive datasets become available.”
Comment 2: Missing Value Imputation Justification: In the data preprocessing step, missing values for expression data (e.g., gene, miRNA, DNA methylation, protein) are simply set to 0 (zero-imputation), while SNP and CNV data are left unimputed. Have the authors evaluated the potential bias introduced by this simplistic zero-imputation method on model performance and subsequent biomarker identification? Please provide the rationale for choosing zero-imputation over more sophisticated imputation techniques (e.g., KNN or mean imputation).
Response 2: We appreciate the reviewer’s valuable comment. Currently, G2PDeep-V2 does not include a built-in data imputation module, and users are expected to perform data imputation externally if desired. In this work, our primary focus is to demonstrate the overall predictive capability and usability of G2PDeep-V2 rather than to compare the effects of different imputation strategies. We therefore applied a simple zero-imputation approach to maintain data consistency across omics types. Notably, even with this straightforward imputation method, the model achieved strong predictive performance, indicating that G2PDeep-V2 is robust to missing values. In future versions, we plan to integrate more sophisticated imputation techniques, such as KNN or mean imputation, to further enhance flexibility and performance (as shown below).
“In the current version, cross-species analyses are not included. We plan to implement a new function for cross-organism validation, accompanied by a comprehensive evaluation to rigorously assess its performance. We also plan to incorporate advanced deep learning interpretability methods, such as Grad-CAM, Layer-wise Relevance Propagation (LRP), and SHAP-based analyses. In future versions, we plan to integrate more sophisticated imputation techniques, such as KNN or mean imputation, to further enhance flexibility and performance.”
Comment 3: Biomarker Interpretation Robustness: The manuscript states that the Multi-CNN model uses the Saliency Map algorithm to identify important biomarkers. Given the known sensitivity of Saliency Maps to small input perturbations, have the authors attempted or considered other more robust or advanced deep-learning interpretability methods (such as Grad-CAM, LRP, or SHAP-based analysis)? A justification for using Saliency Map as the primary method for multi-omics biomarker discovery should be provided.
Response 3: Thank you for pointing it out. In our previous work, we demonstrated that the saliency map algorithm can effectively identify SNPs highly associated with phenotypes. Therefore, we did not repeat this experiment in the current study. We have updated the manuscript accordingly to clarify this point. In future work, we plan to incorporate more advanced deep learning interpretability methods, such as Grad-CAM, Layer-wise Relevance Propagation (LRP), and SHAP-based analyses, as mentioned below and in the Discussion section.
“The significant biomarkers associated with phenotypes of interest to researchers are estimated using models in G2PDeep-v2. Based on our previous work (Liu et al., 2019), the saliency map algorithm is applied to the multi-CNN to identify SNPs highly associated with the phenotype, while the coefficients of traditional ML models are also utilized to pinpoint significant biomarkers.”
“We plan to implement a new function for cross-organism validation, accompanied by a comprehensive evaluation to rigorously assess its performance. We also plan to incorporate advanced deep learning interpretability methods, such as Grad-CAM, Layer-wise Relevance Propagation (LRP), and SHAP-based analyses. In future versions, we plan to integrate more sophisticated imputation techniques, such as KNN or mean imputation, to further enhance flexibility and performance”
Comment 4: Performance Metrics and Computational Resources: The manuscript mentions that training a typical model with automated hyperparameter tuning takes about 2 hours and only utilizes CPU resources. Given the general computational demands of deep learning, does G2PDeep-V2 support the use of GPUs for model training to accelerate tasks? Furthermore, the Results section currently lacks quantitative performance evaluations (e.g., AUC, R-squared, or PCC) for the 23 TCGA cancer studies and the SCN case study. Please supplement the manuscript with a table clearly presenting the key prediction performance metrics for the model in these case studies under different omics combinations.
Response 4: Thank you for the suggestion. Yes, G2PDeep-V2 supports GPU acceleration for model training. We have deployed G2PDeep-V2 on a GPU-enabled server, which significantly reduces computation time compared with CPU-only training (updated in the Discussion section as shown below). To provide quantitative performance evaluations, we have added Supplementary Table S3, which presents the AUC values for all 23 TCGA cancer studies. This table includes key prediction performance metrics across different omics combinations, offering a clear overview of the model’s predictive performance.
“G2PDeep-v2 webserver is developed as a one-stop-shop platform that addresses the need for efficient and accurate phenotype predictions from multi-omics data with customizable deep learning and machine learning models for any organisms. G2PDeep-v2 is the first web server that allows models to be created, trained with automated hyperparameter tuning, and used for inference on multi-omics data uploaded by researchers. We have deployed G2PDeep-v2 on a server equipped with both CPU and GPU resources to expedite model training and inference processes. Performance, compatibility, usability, and interpretability are all central principles of G2PDeep-v2.”
This manuscript is a resubmission of an earlier submission. The following is a list of the peer review reports and author responses from that submission.
Round 1
Reviewer 1 Report
Comments and Suggestions for Authors
The manuscript primarily presents G2PDeep-v2, a web-based deep learning platform designed for phenotype prediction and biomarker discovery using multi-omics data. While it underscores the challenges of integrating multi-omics and outlines platform features such as multi-omics input support, automated hyperparameter tuning, high-performance computing, visualization and downstream gene set enrichment analysis, the work still requires substantial revision.
- The manuscript positions G2PDeep-v2 as “the first web-based deep-learning framework for multi-omics phenotype prediction across all organisms,” but this claim is overstated. Several tools already exist that integrate multi-omics with deep learning (e.g., MOGONET, DeepMO, MOLI, etc.). The innovation here appears more in platform engineering (combining features into a web server) than in methodological breakthroughs.
- The deep learning models (CNN, logistic regression, SVM, random forest, etc.) and Bayesian hyperparameter tuning are not new. Packaging them into a server is useful, but not a true scientific innovation unless there are unique algorithmic contributions.
- While the system accepts multi-omics from plants, animals, and microbes, the manuscript does not show meaningful cross-organism validation. Without such results, the claim of generalizability is weak.
- The introduction emphasizes the challenges of integrating multi-omics data, interpretability, and biomarker discovery. However, the results mostly showcase platform features (uploading datasets, training models, visualization) instead of new insights into biology or model performance benchmarks.
- The manuscript suggests G2PDeep-v2 works for “all organisms,” but the main demonstration is limited to TCGA cancer datasets and soybean. The logical leap from limited demonstrations to universal applicability is not justified.
- The manuscript repeatedly emphasizes the same points (e.g., hyperparameter tuning automation, web-based interface, GSEA integration)
Reviewer 2 Report
Comments and Suggestions for Authors
Zeng et al. developed a web-server, G2PDeep-v2, to implement deep learning for trait prediction and biomarker detection using multi-omics data. This is an updated version based on previously released v1 to incorporate measurements from multiple platforms.
Biomarker discovery, or variable selection, is an essential component of the server. However, it is not clear to users whether variable selection results are valid. According to their paper, machine learning approaches such as logistic regression were included for variable selection. Regression coefficients obtained after model fitting were used to select features, which is potentially problematic to perform selection. It was also mentioned that features with larger coefficients are selected, but this is not quite right when you selected a feature with a coefficient of 0.2 rather than one with coefficient -2. In published work especially with the goal of multi-omics integration, variable selection methods have been extensively developed and applied (Wu et al. 2019). Then, compared to the substantial amount of work in this area, how to justify the feature selection implemented with the server?
According to Wu et al. (2019), integration methods can be categorized as hierarchical and parallel integration, respectively. It appears to me that the server adopts parallel integration, so what matters is the scale or size of the data, since omics measurements from different platforms are treated equally. In this case, I assume that even if users misclassify the data type, for example, inputting gene expression data as CNV, or vice versa, the server will still produce the same results. Alternatively, if users input both gene expression and CNV data together as gene expression data, will that affect the results or trigger any error messages? Could you please clarify?
Another question to address is robustness of the server. Heterogeneity of complex disease demands robust methods. Please refer to Wu et al. (2019) on examples and discussions of robust integration methods. Please provide more details on how this server deals with data heterogeneity, skewed distributions and outliers in phenotype. In particular, how are the prediction performance assessed? Since you work with survival outcomes, can this server generate classical metrics for evaluating the prediction performance of survival models?
In texts related to variable selection, “significant” biomarkers have been frequently mentioned. But picking up a feature with non-zero estimates does not yield “significant” features. This may be confusing to users. I am also wondering about how the server handles SNP data, specifically, whether and how it can directly identify SNPs from specific genes. According to the description, the server works with identified genes, but SNP data typically do not include gene annotations directly.
It may be overly confident to claim that this server works well for all types of data (human, plants, animals, etc.), as each data type has its own unique characteristics and implications.
References:
Wu, C., Zhou, F., Ren, J., Li, X., Jiang, Y., & Ma, S. (2019). A selective review of multi-level omics data integration using variable selection. High-throughput, 8(1), 4.
Reviewer 3 Report
Comments and Suggestions for Authors
The manuscript describes a bioinformatics tool, specifically an advanced version of an existing tool, suitable for analyzing multi-omics data.
My view is that the journal is not the right choice for this type of article. The focus stated on the journal's website is "The journal has a strong focus on structures and functions of bioactive and biogenic substances, molecular mechanisms with biological and medical implications as well as biomaterials and their applications."
The submission under the bioinformatics section is probably insufficient, because the application to biological data is only a marginal part of the article, which is clearly aimed and designed to present the software tool and not advances in the knowledge of molecular mechanisms. This is also evident from the absence of references to the data used for the case studies in the materials and methods section, which is devoted exclusively to describing the software.
Reviewer 4 Report
Comments and Suggestions for Authors
In the present work the authors provided a collection of multi-omics datasets from 23 different cancer types, each labeled with long-term survival data. To address previous limitations, they enhanced the previous platform into G2PDeep-v2—a robust, web-based system for phenotype prediction and biomarker identification and trained deep learning models with automated hyperparameter tuning across various combinations of omics data, successfully identifying multiple sets of key biomarkers. The work is interesting and well organized; therefore, I recommend its acceptance, if the following minor comments can be appropriately addressed.
The overview of the new version is well depicted.
Table 1 is very useful showing that the updated version supports multiple multi-omics data types, offers a wider range of model configurations, and includes advanced hyperparameter tuning options, integrating also Gene Set Enrichment Analysis.
Figure 6 description is mostly clear but needs refinement. Label E is missing, which affects flow. Some terms are too general, such as Figure of multi-CNN model. Moreover, key details are missing, such as the type of learning curves or visualizations used. Dataset context (validation vs. test) could be specified for performance plots like the ROC curve. Check also for minor grammar issues (automate should be automated) or more precise technical terms (model details could be model summary).
Regarding TCGA studies, do you believe the approach has some limitations related to data and combinations? Non-uniform datasets often lack certain omics types, limiting the exploration of all multi-omics combinations and the small number of uniform datasets may reduce the generalizability of findings. Did your research include missing data which can decrease statistical power and/or make it harder to train models that perform well across both ideal and real-world conditions?
Figure 8 and models results could be better organized and illustrated.
Regarding your future directions, which have been discussed extensively, such as meta-learning techniques that allow models to learn from small amounts of data or deploying the updated version on a server, do you believe that the results of the present work are sufficient or not to draw strong conclusions?